# Significantly wetter or drier future conditions for one to two thirds of the world's population

Ralph Trancoso [1,2] ✉, Jozef Syktus [1], Richard P. Allan [3], Jacky Croke [4], Ove Hoegh-Guldberg [1] & Robin Chadwick [5,6]

Future projections of precipitation are uncertain, hampering effective climate adaptation strategies globally. Our understanding of changes across multiple climate model simulations under a warmer climate is limited by this lack of coherence across models. Here, we address this challenge introducing an approach that detects agreement in drier and wetter conditions by evaluating continuous 120-year time-series with trends, across 146 Global Climate Model (GCM) runs and two elevated greenhouse gas (GHG) emissions scenarios. We show the hotspots of future drier and wetter conditions, including regions already experiencing water scarcity or excess. These patterns are projected to impact a significant portion of the global population, with approximately 3 billion people (38% of the world's current population) affected under an intermediate emissions scenario and 5 billion people (66% of the world population) under a high emissions scenario by the century's end (or 35-61% using projections of future population). We undertake a country- and state-level analysis quantifying the population exposed to significant changes in precipitation regimes, offering a robust framework for assessing multiple climate projections.

Precipitation as the primary freshwater source, plays a crucial role in Earth's water availability[1,2]. Understanding changes in precipitation patterns under elevated greenhouse gas (GHG) emissions and their impact on the global population is essential for developing effective adaptation and mitigation strategies and preparing for increased natural disasters[3–5]. While temperature projections show general agreement, precipitation projections exhibit significant regional uncertainties[6–9] and lack coherence among climate models[10].

Precipitation is complex to simulate due to various influencing factors, including diverse physics represented by global climate models (GCMs)[11], their sensitivity to radiative forcing, rate of warming[12,13] and to aerosols radiative cooling[14,15]. Sea surface temperature (SST) also plays a pivotal role in precipitation variability[16] with a series of regional water deficits[17,18] and surpluses[2,19,20] often associated with specific SST patterns[21]. Unforced, internal climate fluctuations operating at timescales varying from intra-seasonal to multi-decadal additionally contribute to precipitation variability, e.g., the El Niño-Southern Oscillation[22], the Indian Ocean Dipole[23], the Pacific Decadal Oscillation[24], the Southern[25] and Northern[26] Annular Modes, and the Atlantic Multi-decadal Oscillation[27]. Future GCM projections of multiple climate modes, their interactions, and resultant teleconnections with precipitation do not align over time across GCMs, amplifying the heterogeneity of projections[6,7]. As climate modelling science progresses, the number of ensembles—i.e., climate

¹School of The Environment, The University of Queensland, Brisbane, QLD, Australia. ²Climate Projections and Services, Department of Environment and Science, Queensland Government, Brisbane, QLD, Australia. ³Department of Meteorology and National Centre for Earth Observation, University of Reading, Reading, UK. ⁴Centre for Climate, Environment and Sustainability, School of Earth and Atmospheric Sciences, Queensland University of Technology, Brisbane, QLD, Australia. ⁵Met Office Hadley Centre, Exeter, UK. ⁶Global Systems Institute, Department of Mathematics, University of Exeter, Exeter, UK. ✉e-mail: r.trancoso@uq.edu.au

modelling experiments representative of physical processes, scenarios, and internally generated climate variability—increases along with computational power. This, however, expands the spread of the climate change signal of precipitation and increases uncertainty. To reconcile the wide range of precipitation projections from multiple GCMs, new approaches are needed[28,29].

Temporal aggregations, although useful for most climate metrics, are inadequate for heterogeneous variables like precipitation. Excessive temporal averaging (e.g., 10+ years) does not retain critical information and may obscure insights into the projected direction of significant changes. To address this issue, we present a novel approach that analyses trends in continuous, long-term time-series[30,31] from multiple GCM ensembles and quantifies the agreement of wetter or drier conditions in terms of precipitation. Our study aims to detect global warming-induced wetting and drying patterns, understand differences between GCM generations, determine seasonal dominance, and identify "hotspots" of drier and wetter conditions with potential global human impacts.

While drier and wetter conditions can also have broader definitions associated with atmospheric demand and characteristics of precipitation[32], here we focus on precipitation totals alone, which is the most important component and with greater uncertainty. We define global warming-induced drying or wetting as statistically significant and substantial continuous decreases or increases in precipitation capable of altering local regimes under intermediate and high emissions scenarios. By using non-parametric trends[30,31] and considering an ensemble of 146 CMIP5[33] and CMIP6[34] climate model runs (Tables S1 and S2), we identify regions where wetting and drying patterns converge across the globe. Our trend-based approach aligns with the continuous nature of radiative forcing[35], provides flexibility and robustness in detecting and quantifying global warming-induced changes, and effectively controls for natural variability[36].

Our innovative approach (see "Methods") evaluates the entire time-series using no interannual averaging and combines information from the fullest range of GCM projections available to determine their agreement and the extent of precipitation impacts with no ensemble aggregation. This is important because each set of simulations provides a plausible storyline of future precipitation patterns under elevated global warming, rather than using aggregated data which can obscure important trends and patterns. The approach also offers an impact-based framework, with country-scale analysis of the drying and wetting agreement, the magnitude of change, and the exposed population, now available to inform climate adaptation policies globally.

Our study advances the understanding of how increased GHG emissions are likely to affect precipitation regimes and impact populations globally[37], providing valuable insights into the direction of precipitation changes under different emissions scenarios and enhancing our ability to develop effective adaptation strategies. By addressing the lack of coherence in precipitation projections, our research contributes to a more comprehensive understanding of the future impacts of climate change on water resources.

## Results

### Wetter and drier conditions across countries and impacted population

Clear hotspots of both wetting and drying agreement emerged from the analysis of 146 GCM runs, highlighting regions where models agree in the direction, magnitude, and statistical significance of trends in long-term simulated precipitation under intermediate and high emissions (Fig. 1).

At the scale of individual countries, Greece, Spain, Palestine, Portugal, and Morocco had a substantial agreement in drying, where at least 85% of the models had robust decreasing trends with a median

potential change in rainfall regimes up to −21% under intermediate emissions and −55% under very high emissions (e.g., in Trinidad and Tobago, Morocco, Grenada, and Gibraltar). In contrast, Finland, North Korea, Russia, and Canada displayed over 90% of agreement in wetting, with some countries exhibiting a cumulative change in the annual regime of over +35% under intermediate emissions and over +48% under high emissions by the end of this century (i.e., in Greenland, Svalbard, Nauru and Kiribati—see Fig. 1a, b and Supplementary Data 1). Countries with higher model agreement on the sign of change also tend to have larger changes projected by the model ensemble. The results also depict the regions where no substantial agreement was observed at the annual scale, such as central Europe, Southwest Asia, Australia, and parts of the African west coast and South America. Some of these regions are subject to significant seasonal changes that may not be apparent at an annual time scale (e.g., Central Europe in JJA and South America in SON; Figs. S1–2 and Supplementary Data 1).

Although the model agreement was greater for the very high emissions scenarios, changes from intermediate to very high emissions were more pronounced for drying than for wetting agreement. The Caribbean (e.g., Cuba, Dominican Republic, and Haiti in Fig. 1c, d) and the Mediterranean (e.g., Greece, Spain, Turkey, and Italy) regions are the areas most impacted by the expansion of the drying zone in terms of both agreement level and magnitude of change from intermediate to very high emissions. These findings highlight the global hotspots of changes in precipitation totals should emissions increase from intermediate to very high levels.

Importantly, for some countries such as the United States, Brazil, Chile, Indonesia, and South Africa with heterogeneous spatial patterns and/or notable internal gradients of wetting and drying agreement, spatial means may be too generalised to inform policies (vertical lines over Fig. 1c, d represent intra-country spatial variability). Therefore, for countries with high internal heterogeneity and for information on more refined spatial scales, decision-makers are recommended to refer to the state-level regionalisations (Fig. 1e–i and Supplementary Data 2) as well as the gridded dataset.

Results suggest that these wetter and drier patterns due to elevated GHGs and global warming will largely affect the countries already experiencing climate change impacts[38,39] (Fig. 2). When a majority threshold is applied (i.e., 50% or 1/2 of simulations), the results indicate that 38% of the current global population[40] are projected to be affected by significant changes in rainfall over this century under intermediate GHG emissions scenario and 65.6% for the very high emissions scenario. This represents 266,52 million people (3.3%) likely to be affected by drier, and 2,76 billion people (34.7%) by wetter conditions under intermediate GHG emissions (Fig. 2a, c). This increases to almost one billion (875,47 million; 11%) people affected by drying and over four billion (4,35 billion; 54.6%) people by wetting under very high emissions (Fig. 2b, c). When future population projections are considered, 35.5% (3,26 billion people) and 61.4% (4,64 billion people) of the 2100 world's population are projected to be affected by wetter or drier conditions under moderate and very high emissions respectively. The distribution of future affected populations across countries is consistent with our current population[40] estimates (Fig. S3).

When a more conservative agreement threshold is used (i.e., 66% or 2/3 of simulations), the current affected population decreases for intermediate and very high emission scenarios—9.9% (1.2% for drying and 8.7% for wetting) to 42.9% (7% for drying and 35.9% for wetting) respectively. The projected future population[41] impacted ranges from 6.2% (1.0% for drying and 5.2% for wetting) to 39.7% (7.0% for drying and 32.7% for wetting) from intermediate to very high emissions. However, by factoring in trend probabilities, direction and regime changes using the entire time-series, our trends-based approach is more rigorous than previous studies quantifying agreement of projections using long-term averages[42]. Therefore, the majority

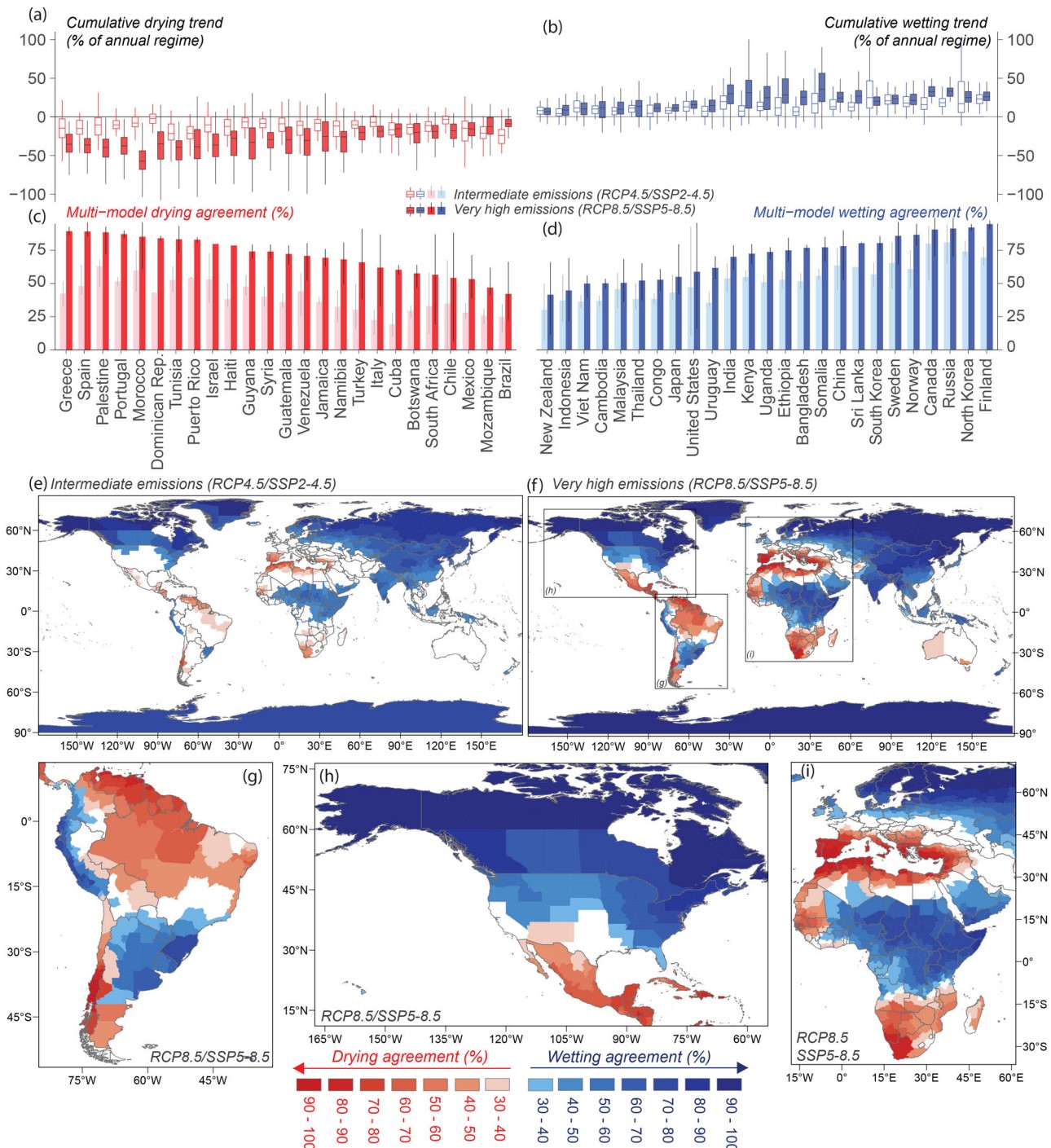

**Fig. 1 | Country- and state-level agreement on drying and wetting patterns under intermediate and very high emissions scenarios detected from an ensemble of 146 CMIP5 and CMIP6 GCMs. a, b** Box-whiskers plots of cumulative change in precipitation regime over a 120-year period across GCMs for selected countries with higher agreement and affected populations. **c, d** Selected countries ranked by drying and wetting multi-model agreement (vertical lines denote intra-country variability showing the 10th and 90th spatial percentiles). **e, f** Spatial distribution of drying and wetting multi-model agreement for states. Rectangles on **f** very high emissions panel focusing on regions with strong intra-country variability are shown in detail on panels for **g** South America, **h** North America, **i** Africa, and Europe. White regions indicate no substantial agreement for drying and wetting. Refer to Figs. S1 and S2 and Supplementary Data for seasonal agreement across all countries and states globally. For refined spatial patterns at grid-cell level refer to Fig. 3.

agreement threshold (50%) is more suitable for the purpose of this study and the 66% agreement threshold is overly conservative and would conceal some important insights on impacts revealed globally. Many populated regions with substantial drying agreement such as the Mediterranean Europe, North Africa, Central America and the Caribbean and southern South America, eastern Brazil, the Amazon, and Western Australia are already facing water scarcity[43] or have experienced severe droughts[39,44] in the recent past. Conversely, highly populated areas over Asia, Northern Europe, north-western United States, and central Africa have a substantial agreement for wetter conditions under moderate and high emissions. Many of these areas experienced major flood and extreme precipitation events in the recent past, such as Bangladesh, India, Japan, and East Africa, and are still recovering from significant losses[38,39,45–47].

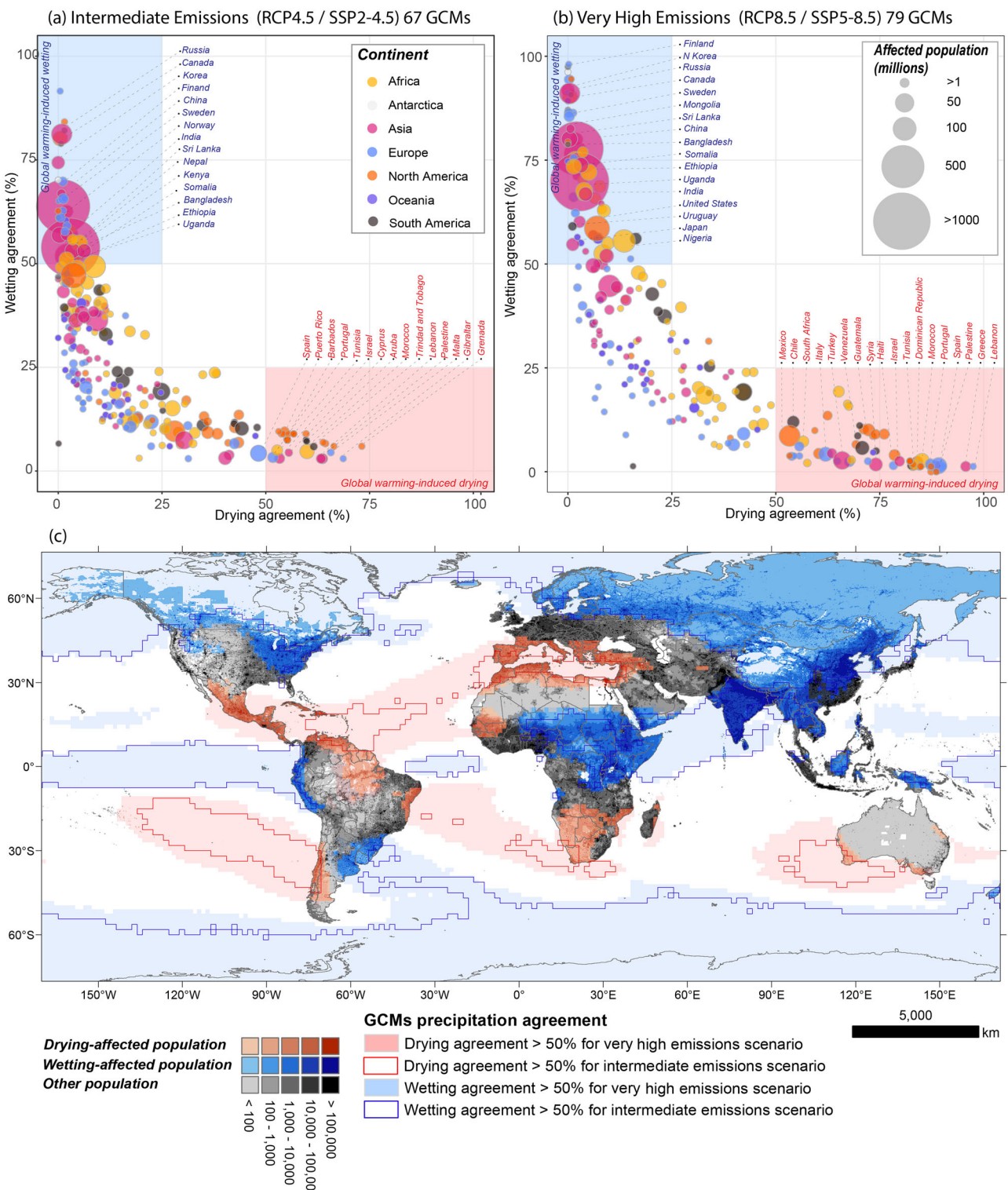

**Fig. 2 | Current population affected by global warming-induced wetting and drying based on the agreement of long-term annual precipitation trends of 146 CMIP5 and CMIP6 GCMs.** Country-level regionalisation of drying and wetting agreement and affected population forced under **a** intermediate and **b** very high emissions. Colours denote continents and bubble size shows the affected populations by changing precipitation. Red and blue rectangles indicate the regions where the majority of GCMs agree. Countries with higher agreement, population, and representatives of different regions are labelled. Refer to Fig. S3 for scatterplots using future population projections. **c** Spatial distribution of regions with global warming-induced drying and wetting (agreement >50%) under intermediate and very high emissions and affected population. Contour and shaded (red and blue) areas refer to intermediate and very high emission scenarios. Refer to Supplementary Data 1 and 2 for the population affected by global warming-induced wetting and drying across all countries.

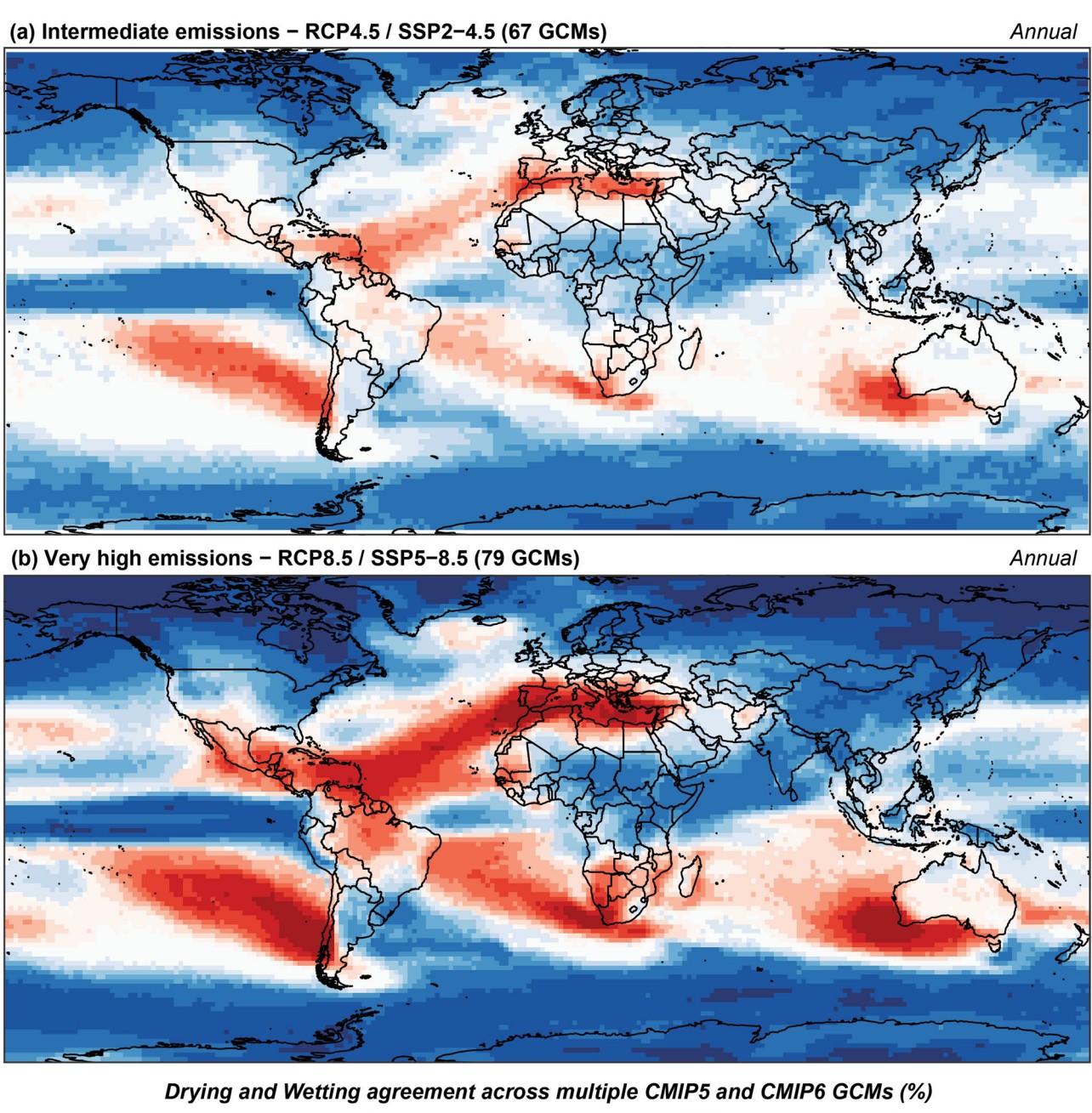

**(a) Intermediate emissions − RCP4.5 / SSP2−4.5 (67 GCMs)** *Annual*

**(b) Very high emissions − RCP8.5 / SSP5−8.5 (79 GCMs)** *Annual*

*Drying and Wetting agreement across multiple CMIP5 and CMIP6 GCMs (%)*

100  90  80  70  60  50  40  30  20    20  30  40  50  60  70  80  90  100

**Fig. 3 | Multi-model (CMIP5 and CMIP6) drying and wetting agreement of robust trends in annual precipitation. a** Intermediate (67 GCM runs) and **b** high emissions (79 GCM runs). Shades of red denote drying agreement and shades of blue indicate wetting agreement.

## Seasonal wetting and drying agreement and dominance

Separate assessments of CMIP5 and CMIP6 GCMs across different calendar seasons and emissions scenarios demonstrated a substantial overlap in the hotspots of wetting and drying GCMs agreement (Figs. S4 to S7) with, in some cases, up to 100% of agreement amongst the entire ensemble of 146 GCMs used. Overall, CMIP5 and CMIP6 models yielded very similar agreement for annual precipitation totals and across calendar seasons. However, CMIP5 models generally had lower drying agreement in the southern hemisphere and higher in the northern hemisphere, except for JJA and SON where CMIP5 had greater drying agreement (Figs. S4 and 5). CMIP6 wetting was greater across most of the calendar seasons, especially across high latitudes (Figs. S6 and 7). This is likely to be linked to the higher equilibrium climate sensitivity of CMIP6 models[12,13].

Given this similarity, both CMIP5 and CMIP6 models were analysed together for intermediate and high emission scenarios (Figs. 3 and 4).

The seasonal wetting and drying patterns of future precipitation show a sharp drying-to-wetting gradient from south to north-west of the United States (US) for MAM precipitation (Fig. 4f). Interestingly, the wetting-to-drying gradients within the US were inverted from MAM to JJA (Fig. 4f, g). Similar gradients are observed over Africa with wetting trends over the north and drying over the south in all seasons (Fig. 4). A strong JJA and SON drying is detected over south-western Australia and north-eastern Brazil (Fig. 4g, h). These spatial patterns of wetting and drying are consistent with previous studies assessing multiple GCMs[42,48–50], but none of them evaluated continuous time-series and used the fullest GCM data resource to encapsulate the

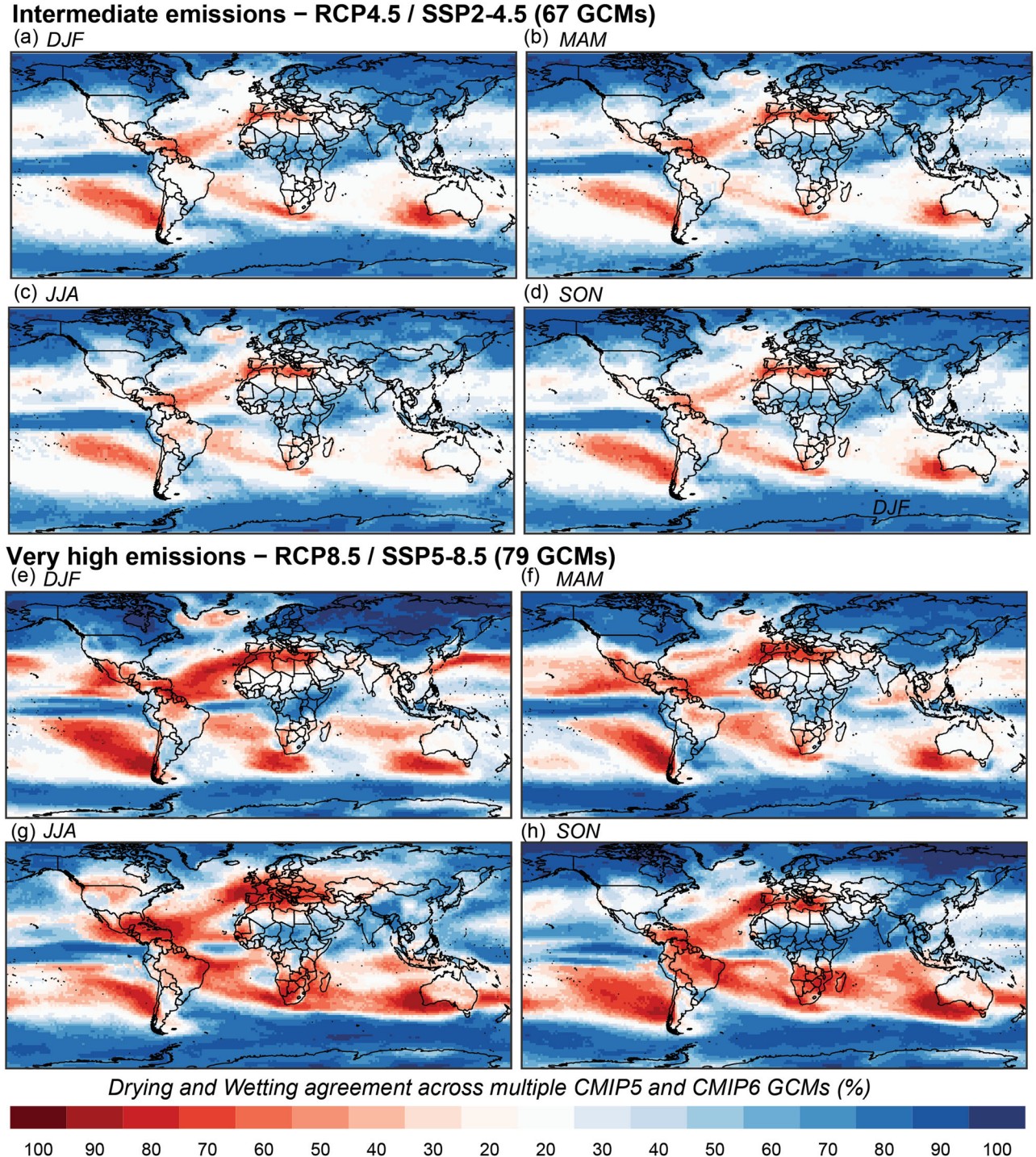

**Intermediate emissions – RCP4.5 / SSP2-4.5 (67 GCMs)**
(a) *DJF*  (b) *MAM*  (c) *JJA*  (d) *SON*

**Very high emissions – RCP8.5 / SSP5-8.5 (79 GCMs)**
(e) *DJF*  (f) *MAM*  (g) *JJA*  (h) *SON*

*Drying and Wetting agreement across multiple CMIP5 and CMIP6 GCMs (%)*

100 90 80 70 60 50 40 30 20 | 20 30 40 50 60 70 80 90 100

**Fig. 4 | Multi-model (CMIP5 and CMIP6) drying and wetting agreement of robust trends in seasonal precipitation. a–d** Intermediate (67 GCM runs) and **e–h** high emissions (79 GCM runs).

nuances in spatial patterns, model agreement, and impacts over global population showcased in this study. In some European countries, such as the United Kingdom, Germany, Austria, Belgium, and Switzerland, a strong drying agreement was detected in JJA (Fig. 4g) and a wetting agreement in DJF (Fig. 4h). These significant wetting and drying seasonal trends offset the annual scale signals and tend to increase magnitude with radiative forcing (Supplementary Data 1 and 2).

To assess which season contributes the most to the annual wetting and drying, those regions with model agreement greater than 50% at the annual scale were selected. The season with greater trend magnitude is termed the seasonal dominance of wetting and drying

agreement. Results reveal spatially consistent patterns across emission scenarios with no globally dominant calendar season for the detected wetting or drying. However, interesting patterns of where, and on which calendar seasons, long-term trends in regional rainfall are more pronounced are apparent. This has the effect of unravelling simultaneously where long-term precipitation projections from multiple climate models agree and which season is most dominantly driving such alterations in precipitation totals in a warmer world (Fig. 5). For example, the drying over the southwest coast of Australia and the Indian Ocean is dominated by SON precipitation trends, whereas over the Iberian Peninsula, it is dominated by JJA precipitation trends.

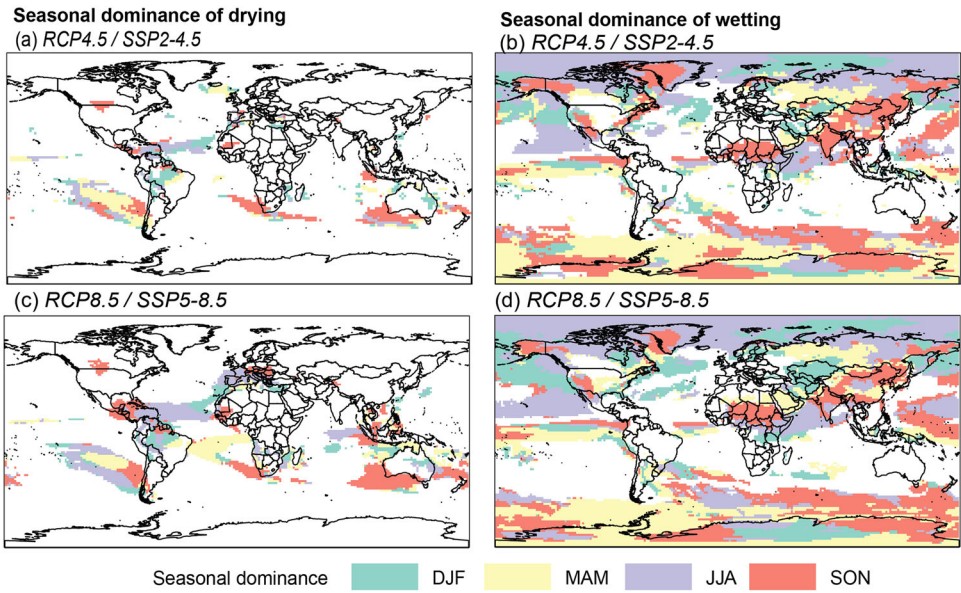

**Fig. 5 | Dominance of calendar seasons over the regions where the majority of GCMs projects significant wetter and drier conditions at annual scale by century's end. a, b** Intermediate (RCP4.5 / SSP2-4.5) and **c, d** very high (RCP8.5 / SSP5-8.5) emissions.

Likewise, SON dominates wetting convergence over China, India, and Central Africa whereas MAM is the dominant driver in the United States. The analysis reveals insights into how future changes in precipitation may manifest across calendar seasons. In central Africa, for instance, the season that dominates wetting varies with latitude with no clear progression over the year whereas over northern Eurasia there seems to be an eastward seasonal progression. In the Amazon, the dominance of the seasons on drying has complex patterns with DJF and JJA dominating the south and SON the north.

## Discussion

This study estimates the extent of the current global population to be affected by significant long-term changes in precipitation due to human-caused global warming. The analysis provides an intermediate to high emissions envelope for how the global population is projected to be impacted by future changes in long-term precipitation totals based on the agreement of precipitation projections from multiple state-of-the-art climate models. A robust multi-model agreement of drier conditions was detected over mid-latitudes in the Mediterranean region, southern Africa, Central America, and South America. Likewise, we found a substantial multi-model wetting agreement over high latitudes, such as the Scandinavian region and northern North America, and in other regions such as most of Asia and Central Africa. Several global-scale controls influence changes in the atmospheric circulation, such as Hadley Cell expansion[51], weakening of the Walker circulation[52], changes in the intertropical convergence zone[53], warming contrasts between land and ocean[54], poleward precipitation transport from warm to cold ocean regions and vegetation response to higher atmospheric $CO_2$[55,56] have also been connected to drier conditions over mid-latitudes and wetter conditions over low and high latitudes[51,57]. Model deficiencies such as ITCZ biases[58–60] might also be affecting the drying patterns across specific regions—e.g., the Amazon and parts of Northeast Brazil.

This is a significant contribution towards advancing our understanding on what is the expected signs of changes in precipitation and how it is projected to change globally under intermediate to very high emissions. The regions subject to robust wetting and drying cover a vast fraction of the global land, which means that 38% of the current world's population, or 3 billion people would be affected by substantially changing precipitation amounts by the end of this century

should the world experience intermediate GHG emissions while 65.6% of the world's population or 5 billion people would be affected in a very high GHG emissions scenario. Projections of future population[41] yield consistent results for the affected population range (35.5–61.4% of the world's population). The study also highlights the regions where limited model agreement on change was registered, such as central Europe, Southwest Asia, Australia, and parts of the African west coast and South America. To the best of our knowledge, no previous research was able to encapsulate agreement of future wetting and drying and the affected population at a country-scale using all the latest GCMs with no multi-ensemble and interannual aggregations. This is important because different models represent distinct and physically plausible storylines of future precipitation under elevated greenhouse gas levels, which is a valuable data resource to be explored altogether rather than aggregated. While many GCM runs are used for each scenario, the diversity of these runs is lower than the total number might suggest as there are multiple runs of similar GCMs. However, when a subset sampling is applied to account for independence following the model genealogy[11], the hotspots of wetting and drying patterns are consistent.

Changing patterns in totals and extremes[61–63] and increasing contrast between dry and wet precipitation seasons[64,65] have been previously linked to global warming and regional patterns of sea surface temperature[21,66,67]. Extreme droughts and precipitation events over the past two decades were also connected to observed increasing temperature[39], revealing observational spatial patterns of changing precipitation consistent with our findings for future projections. Several regional characteristics of changes in seasonal future precipitation have been suggested as drivers across the regions highlighted in this study as subject to wetter and drier conditions. They include longer dry spells and lower dry season precipitation in the Amazon, Central America, and south of the African continent and less precipitation over the Iberian peninsula in the wet season driving drying patterns and increased dry and wet season precipitation in north-west Africa and southeast Asia driving wetting patterns[49]. This study has focused on how the total amounts across seasons are projected to change and how different GCMs from distinct generations agree under intermediate and high emissions. It is worth noting that changes in total amount (i.e., volume) are likely to be a consequence of changing intensity, frequency, and duration aggregated over seasons. Likewise,

future wetter and drier conditions will play a big role favouring the conditions conducive to increasing extreme events such as floods and wildfires, thereby harsher extreme events are also expected to affect the population globally across the regions highlighted in this study. If a sharp reduction of emissions is adopted, the extent of significant drier conditions and impacted population may be reduced across Central America, the Mediterranean region, and south of the African continent, but the spatial extent of substantially wetter conditions (i.e., precipitation totals) and their impacts may be less affected by limiting emissions (Fig. S8).

The approach detected agreement across multiple models in future wetting and drying trends, revealing critical information on how precipitation is projected to change under scenarios associated with continued GHG emissions. By examining the time-series of individual models with flexible trend detection methods, the approach provides a more robust quantification of change, summarising critical multi-model information. This is a more comprehensive way to sample climate projections of heterogeneous variables and analyse multiple GCMs and provides a substantial advance over previous studies based on comparison of long-term means or multi-model averaging. A further innovation is the quantification of precipitation changes at country-scale and their potentially exposed populations. These findings can directly assist with designing 'fit for purpose' climate adaptation policies and reduce uncertainty in which direction precipitation is projected to change globally under different emissions levels.

## Methods

### Climate model simulations
Our analysis included 146 GCMs, 67 for intermediate GHG emissions scenarios and 79 for very high emissions scenarios: 25 CMIP5 forced under RCP4.5, 44 CMIP6 forced under SSP2-4.5 (Table S1), 35 CMIP5 forced under RCP8.5 and 44 CMIP6 forced under SSP5-85 (Table S2)[33,34]. The data was sourced from CMIP data portal (https://esgf-node.llnl.gov/search/cmip5/ and https://esgf-node.llnl.gov/search/cmip6/). To understand how projected precipitation from multiple GCMs respond to warming over this century, we used simulations forced with both intermediate and very high emissions. By focusing on both intermediate and very high emission scenarios, which constitute a plausible upper bound without substantial mitigation of future radiative forcing, this contribution targets the likely envelope of precipitation changes and their impacts. While it has been argued that very high emission scenarios are less plausible given the current policies to reduce emissions, this contribution focuses specifically on what is the expected direction of a significant change in precipitation totals under intermediate and very high emissions, and what is the resultant impact envelope. We also included the high climate sensitivity models[12] as for the purpose of this analysis, each realisation offers additional insight into physical processes and the opportunity to explore the parameter space.

### Statistical methods for detection of wetter and drier conditions
We explored seasonal time-series from 1980 to 2099 (120 years) with non-parametric trends using the Mann-Kendall test[30] to detect trends and statistical significance and the Theil-Sen Slope[31] to assess trend magnitude. A key advantage of these tests is that they capture the long-term monotonic trend over the entire time-series by looking at how each data point behaved in comparison to the previous data point, which is a more robust estimate of change than comparing two 20-year periods averaged. For this reason, the approach is not sensitive to the start and end points or outliers in the time-series. Another advantage of using trends over a long time-period is constraining natural, internally generated variability of precipitation, as a 120-year period is considered long enough to encompass multiple oscillations of the drivers of global rainfall

(e.g., ENSO, IPO and IOD) rather than aggregating positive or negative phases of these modes. This is important in reducing their influence on the climate change signal.

### Multi-model agreement
A metric to assess drying and wetting agreement within multiple GCMs is proposed to quantify the percentage of GCMs with a robust long-term drying and wetting signal. The metric first assesses the time-series of individual models and then produces an integrated multi-model agreement quantification. It utilised 120-year time-series of annual and seasonal precipitation totals of the 146 GCMs, interrogating the time-series of individual grid-cells as follows:
(i) Whether statistically significant trends ($p < 0.05$) have been detected;
(ii) The direction of trends (slope) to determine if it was undergoing wetting or drying;
(iii) Whether the cumulative trend over the 120-year period (slope) shifted by at least 10% the local regime (or whether the 120-year change is at least 10% as large as the mean).

For the full dataset of 146 GCMs, the number of models that met the three criteria for wetting and drying on an annual and seasonal basis was calculated to estimate the multi-model agreement at the grid-cell scale. Further details on the implementation in R can be found in https://zenodo.org/record/7960484. We considered two agreement thresholds for the assessment—50% (majority of all model simulations) and 66% (2/3 of all model simulations). No ensemble aggregation is used, and ensemble runs are assessed individually and then summarised as agreement. The approach was repeated for the ensembles of CMIP5 and 6 models as well as the full ensemble across calendar seasons and annually for both emission scenarios.

We also assessed the seasonal dominance of the global drying and wetting under increasing GHG emissions. We asked which season has contributed the most to the annual trends for each individual GCM across the regions within the drying and wetting agreement masks. The seasonal dominance was then obtained from the median of all ensembles.

### Country- and state-scale impacts
The population affected by global warming-induced wetting and drying was estimated using gridded datasets representing both current[40] and future scenarios[41]. The total world population size is projected to be greater than the current population under moderate emissions (SSP2-4.5) and lower than the current population under very high emissions.[68] Then the total population within the dry and wet masks was quantified using the 1 km gridded population datasets. We applied an agreement threshold to define the dry and wet masks. Previous studies based on comparison of long-term averages evaluating smaller ensembles used 66% as a measure of consensus[42]. This study investigates agreement rather than consensus, uses long-term non-parametric trends instead of changes in long-term averages, and across a much larger ensemble size, including all the CMIP5 and CMIP6 models with available data. Stepwise statistics were used to bin the agreement distributions from 0 to 100% at 1% intervals. We found that the likelihood of grid cells above 39% and 34% of drying and wetting respectively to equal or exceed the opposite category tends to zero. We concluded that a majority threshold considering 50% of agreement would be appropriate to produce spatial masks of wetting and drying that would not overlap anywhere. The approach mapped four spatially distinct areas exposed to wetting and drying under intermediate (RCP4.5/SSP2-4.5) and very high (RCP8.5/SSP5-8.5) emissions. These masks were used to estimate the population exposed to wetting and drying patterns on a grid-cell basis. To estimate the global population affected, the 1 km grid cells falling within the wetting and drying agreement masks are summed. To estimate the country- and state-

scale affected populations, the grid cells falling within the agreement and country masks are summed. We produced country- and state-scale statistics, including the seasonal agreement of global warming-induced wetting and drying, the trend slopes, and the affected population under intermediate and high emissions (Supplementary Data 1 and 2). Countries and States boundaries were obtained from a database of global administrative boundaries https://gadm.org/download_world.html.

## Data availability

The GCM precipitation projections datasets are available for the public with no restrictions on the CMIP5 and CMIP6 repositories: https://esgf-node.llnl.gov/search/cmip5/ and https://esgf-node.llnl.gov/search/cmip6/. Countries and States boundaries are available from the global administrative boundaries dataset: https://gadm.org/download_world.html. The gridded datasets produced by this research are available on Figshare[69].

## Code availability

The R code used for the calculation of seasonal trends, model agreement, and seasonal dominance of annual changes is available from *Zenodo*[70].

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

## Acknowledgements

This research was partially funded by the Australian Research Council Discovery Project grant DP160102107. Queensland's Department of Environment and Science provided computational support for our analysis. Nathan Toombs and Sarah Chapman assisted with the download and temporal aggregation of GCMs precipitation data. Andy Pitman, Scott Power, John Carter and Craig Hempel provided comments on earlier versions of the manuscript that contributed to its improvement.

## Author contributions

R.T. conceived the original idea for the study, performed the analysis, created the visualisations, and wrote the first draft of the manuscript. J.S., J.C., R.A., O.H., and R.C. provided additional ideas to improve methodology and analysis and contributed towards editing the manuscript.

## Competing interests

The authors declare no competing interests.
