## [Peer Review File · Nature Communications]

Significantly wetter or drier future conditions for one to two thirds of the world's populationREVIEWER COMMENTS

Reviewer #1 (Remarks to the Author):

Overview: this article discusses future projections of climate changing focusing on precipitation. This study investigates trends (non-parametric methods) across 146 GCMs under intermediary and elevated greenhouse emissions (models from CMIP5 and CMIP6 generations). The main goal is to provide a country-level analysis quantifying the population exposed to significant changes in precipitation regimes. The methodology considers time-series from 1980-2099 and investigates trends at model grid level. Conclusions are based on percent of agreement among models on wet or dry trends. The statistical significance is assessed by non-parametric Mann-Kendell tests, and trends are estimated with the Theil-Sen Slope to estimate the trend magnitude.

The goal of this article is to assess populations that will be subject to wet or dry trends. While the topic is impactful, conclusions at country level are generalized and may lead to misinterpretation of results, an undesirable outcome as the topic is of great importance and the 'takeaway messages' need to be accurately delivered, as such papers can have broad impacts. Thus, I urge authors to carefully consider the way they are conveying their messages and be careful in providing figures that will not lead to misinterpretation and further confusion. I will separate these major issues here

a) comparisons at country level

Nations considered in this discussion have different territory and the distribution of the population vary significantly within these territories. Results presented in the document and supplemental figures clearly show that agreements are not observed over large areas (this is common sense among the science community) and generalizations can lead to false interpretation. Take for example South America and the annual agreement (for simplification). The 'wetting' agreement (S2) is more consistent over subtropical south America (affecting Southern Brazil, Uruguay, NE Argentina and subtropical Atlantic Ocean), whereas the 'dry agreement' is observed over Northern Amazon and adjacent countries. Likewise, North America (Canada, the U.S.) show a clear seasonality in the signal of the agreement with large spatial variability (different seasons revert the signal of agreement in some parts of the U.S. and Canada). Thus, I understand that the idea of Figure 1 is to show a summary of these findings, but the generalizations at country level are not realistic, and may lead to undesired misinterpretations. For instance, Brazil appears at the 40-50% drying agreement whereas only Uruguay shows high agreement (although not catalogued with the bar-plots). The drying agreement was mostly consistent (over seasons) in the Amazon and parts of Northeast Brazil – which might still be resulting from the poor representation of the Atlantic ITCZ, a problem that does not seem to have improved with the CMIP6 generation. Perhaps authors should comment on that too, as they likely have examined seasonal precipitation fields in these models. The question is, how decision makers, the media and the readers of this article will interpret Figure-1 plots? Essentially, that future scenarios of emissions should lead to enhanced droughts in Brazil, Southern and Northern Africa, and enhanced wetting conditions in Canada and the U.S. (to cite the large countries that are discussed here) These assumptions are misleading and can have serious

consequences! If a country's budget is designed to help mitigating the effects of climate change, results presented in an article in Nature can seriously influence how decision makers will plan to deal with the climate crisis and future disasters. Canada has been a good example in 2023, and there is little doubt that we could attribute JJA fire season to the expected drying trends shown in JJA in Fig. S7 in the present article. Decision-makers and the media will focus on Fig. 1, and probably ignore S7. In summary, maps that appear in Figure 1 should be replaced with maps that are more realistic (with the appropriate regionalization of the results (as, for instance, shown in Figure 2).

b) The issue with the population affected by future emissions.

I may have missed some important information about how the population was accounted for in this study, but it was not clear in the provided reference (Tatem, World Pop) and respective links whether any projections of population change was considered here. It seems that population was estimated based on the 2021 census for most countries. The question is, are populations assumed to be constant over time? What was used to estimate future projections? This does not appear in the article (or the supplemental material. Thus, this needs further clarification or at least a caveat indicating that these numbers can change

Regardless of the methodology to account for the trend in population, again, it is unclear how the number "4,350,912,136" people was estimated (how this aggregation was done at grid level, since the methodology to estimate trends was performed at grid level). How sure are we that this number (which is very precise in terms of digits) is not under (or over) estimating the expected affected population? Of particular concern is Figure 2 "warming agreement" where circles represent the affected population. Are these circles proportional to the entire population of a given country or the population effectively affected by model agreements? This needs to be properly discussed in the figure captions (bubble size shows population...).

c) Comments on the literature review and introduction:

"Climate modes such as El Niño-Southern Oscillation, Indian Ocean Dipole, Southern Annular Mode, and Atlantic Multidecadal Oscillation contribute to precipitation variability". The oversimplification here needs attention. It is understandable that not all possible coupled modes of variability that influence precipitation can be discussed here, but the way the sentence is written makes the non-expert reader believe that these are the only existing modes modulating precipitation. Why the Southern Annular mode and not the Northern Annular mode, or why the PNA, the PSA, the NAO and PDO are not mentioned here? I suggest that this sentence be re-written to indicate that there are modes that vary from interannual (actually from intraseasonal) to multi-decadal time scales that influence precipitation, and cite a few of them to exemplify.

"Future projections for climate modes do not align over time across GCMs, resulting in highly heterogeneous projections (ref 6,7)". This sentence seems to imply that, because climate modes (say El Niño) do not align over time across GCMs, there is disagreement among models. This is obviously not correct when considering long periods (the reason why we normally consider more than one decade). Moreover, this is not said in any of the two cited references. So, please rephrase or remove this sentence from the manuscript.

Reviewer #2 (Remarks to the Author):

General Evaluation:

This manuscript aims to advance our understanding of how the continuous increase of greenhouse-gases (GHG) might affect global precipitation regimes and furthermore impact global populations. An ensemble of 146 CMIP5/6 model simulations under intermediate and high emission scenarios is applied to detect agreement in wetter and drier conditions across the world and more interestingly connect precipitation change to exposed population at the country scale. There are already many regions experiencing precipitation regime change at both wet and dry end. This research indicates that this trend will continue and more and more people would be affected by the end of this century. The manuscript is well-organized, and the results are very interesting and have great significance in the field. In particular, the country-scale analysis offers a good framework for practical application such as designing/making climate adaption policies for different regions. I suggest it be accepted for publication in the current format.

Minor typos:

1) Line 227, "county" should be "country"

2) Line 244, add "is" before "worth"?

RESPONSE TO REVIEWERS' COMMENTS ON NATURE COMMUNICATIONS MANUSCRIPT NCOMMS-23-26096

"Significantly wetter or drier future conditions for one to two thirds of the world's population"

ID	COMMENT	RESPONSE	STATUS
Reviewer #1			
R1.1	Overview: this article discusses future projections of climate changing focusing on precipitation. This study investigates trends (non-parametric methods) across 146 GCMs under intermediary and elevated greenhouse emissions (models from CMIP5 and CMIP6 generations). The main goal is to provide a country-level analysis quantifying the population exposed to significant changes in precipitation regimes. The methodology considers time-series from 1980-2099 and investigates trends at model grid level. Conclusions are based on percent of agreement among models on wet or dry trends. The statistical significance is assessed by non-parametric Mann-Kendell tests, and trends are estimated with the Theil-Sen Slope to estimate the trend magnitude. The goal of this article is to assess populations that will be subject to wet or dry trends. While the topic is impactful, conclusions at country level are generalized and may lead to misinterpretation of results, an undesirable outcome as the topic is of great importance and the 'takeaway messages' need to be accurately delivered, as such papers can have broad impacts. Thus, I urge authors to carefully consider the way they are conveying their messages and be careful in providing figures that will not lead to misinterpretation and further confusion. I will separate these major issues here	Thank you very much for the time allocated to review our material and for your in-depth assessment. The valuable points raised certainly contributed to improve the quality of our material. We carefully thought about the way we presented our regionalizations and improved the analysis to account for intra-country variability of model agreement, which we detail below on topic R1.2.	DONE
R1.2	a) comparisons at country level Nations considered in this discussion have different territory and the distribution of the population vary significantly within these territories. Results presented in the document and supplemental figures clearly show that agreements are not observed over large areas (this is common sense among the science community) and generalizations can lead to false interpretation. Take for example	We think this is a valid point, which we carefully addressed using a three-tier strategy to reduce generalizations as follow:  1) Inclusion of spatial variability lines (denoting 10th and 90th spatial percentiles per country) over Figure 1 bar plots showing country-level model agreement; 	DONE

South America and the annual agreement (for simplification). The 'wetting' agreement (S2) is more consistent over subtropical south America (affecting Southern Brazil, Uruguay, NE Argentina and subtropical Atlantic Ocean), whereas the 'dry agreement' is observed over Northern Amazon and adjacent countries. Likewise, North America (Canada, the U.S.) show a clear seasonality in the signal of the agreement with large spatial variability (different seasons revert the signal of agreement in some parts of the U.S. and Canada). Thus, I understand that the idea of Figure 1 is to show a summary of these findings, but the generalizations at country level are not realistic, and may lead to undesired misinterpretations. For instance, Brazil appears at the 40-50% drying agreement whereas only Uruguay shows high agreement (although not catalogued with the bar-plots). The drying agreement was mostly consistent (over seasons) in the Amazon and parts of Northeast Brazil – which might still be resulting from the poor representation of the Atlantic ITCZ, a problem that does not seem to have improved with the CMIP6 generation. Perhaps authors should comment on that too, as they likely have examined seasonal precipitation fields in these models.

- 2) Calculations of regional statistics of model agreement and population exposure across all 3752 states globally and inclusion as supplementary dataset;
- 3) Country-level maps were replaced by state-level maps to show intra-country spatial variability yet aggregated across jurisdictions to facilitate decision making. We also included detailed map insets to better illustrate regions with elevated internal spatial variability and sharp gradients.

In addition, we included a paragraph discussing the matter and encouraging readers to refer to the gridded dataset for refined spatial scales insights and interpretations as follow:

“Importantly, for some countries such as the United States, Brazil, Chile, Indonesia and South Africa with heterogeneous spatial patterns and/or notable internal gradients of wetting and drying agreement, spatial means may be too generalized to inform policies (see lines over Figure 1 bar charts representing spatial variability). Therefore, for regions with high internal heterogeneity as well as for information on more refined spatial scales, decision-makers are recommended to refer to the state level regionalizations instead (Figure 1 (maps) and Table SX) as well as the gridded dataset.”

Uruguay has also been included in the bar plots.

We have also added the following sentence in the discussion:

“Model deficiencies such as ITCZ biases^{56–58} might also be affecting the drying patterns across specific regions – e.g., the Amazon and parts of Northeast Brazil.”

		References Hwang, Y.-T. & Frierson, D. M. W. Link between the double-Intertropical Convergence Zone problem and cloud biases over the Southern Ocean. Proc. Natl. Acad. Sci. 110, 4935–4940 (2013). 57. Mamalakis, A. et al. Zonally contrasting shifts of the tropical rain belt in response to climate change. Nat. Clim. Chang. 11, 143–151 (2021). 58. Tian, B. & Dong, X. The Double-ITCZ Bias in CMIP3, CMIP5, and CMIP6 Models Based on Annual Mean Precipitation. Geophys. Res. Lett. 47, e2020GL087232 (2020).	
R1.3	The question is, how decision makers, the media and the readers of this article will interpret Figure-1 plots? Essentially, that future scenarios of emissions should lead to enhanced droughts in Brazil, Southern and Northern Africa, and enhanced wetting conditions in Canada and the U.S. (to cite the large countries that are discussed here) These assumptions are misleading and can have serious consequences! If a country's budget is designed to help mitigating the effects of climate change, results presented in an article in Nature can seriously influence how decision makers will plan to deal with the climate crisis and future disasters. Canada has been a good example in 2023, and there is little doubt that we could attribute JJA fire season to the expected drying trends shown in JJA in Fig. S7 in the present article. Decision-makers and the media will focus on Fig. 1, and probably ignore S7. In summary, maps that appear in Figure 1 should be replaced with maps that are more realistic (with the appropriate regionalization of the results (as, for instance, shown in Figure 2).	Figure 1 has been redesigned to address these points following detailed response on topic R1.2 and new state-level regionalizations have been presented as maps and included as supplementary data with more realistic regionalizations for decisionmakers. Figure S7 (now Figure S8 in our revised version) shows differences in drying and wetting agreement from intermediate to very high-emission scenarios – that is [RCP8.5 and SSP5-8.5] – [RCP4.5 and SSP2-4.5] for annual and seasonal rainfall. It is insightful but its interpretation is complex, and we thought it could confuse the general public. That's why we decided to include it as a supplementary figure.	DONE
R1.4	b) The issue with the population affected by future emissions. I may have missed some important information about how the population was accounted for in this study, but it was not clear in the	Thanks, this is another important point that we are happy to be provided with the opportunity to address.	DONE

provided reference (Tatem, World Pop) and respective links whether any projections of population change was considered here. It seems that population was estimated based on the 2021 census for most countries. The question is, are populations assumed to be constant over time? What was used to estimate future projections? This does not appear in the article (or the supplemental material. Thus, this needs further clarification or at least a caveat indicating that these numbers can change

In this revised version of our manuscript, we have also estimated the future affected population using downscaled population projections for moderate and high emission scenarios (KC & Lutz 2017; Wang et al 2022).

We have included in the text global statistics for future population under different emissions. A supplementary figure repeating Figure 2 bubble plots with estimated affected populations by country using future population was included as well.

“When future population projections are considered instead, 35.5% (3,26 billion people) and 65.6% (5,22 billion people) of the 2100 world's population are projected to be affected by wetter or drier conditions under moderate and very high emissions respectively. The distribution of future affected population across countries is consistent with our current population estimates (Figure S3).”

In addition, we produced regionalised statistics including both current and projected future population at both country- and state-levels to be released as supplementary dataset with the goal of facilitating climate adaptation and decision-making globally.

Wang, X., Meng, X. & Long, Y. Projecting 1 km-grid population distributions from 2020 to 2100 globally under shared socioeconomic pathways. *Sci. Data* 9, 563 (2022).

KC, S. & Lutz, W. The human core of the shared socioeconomic pathways: Population scenarios by age, sex and level of education for all countries to 2100. *Glob. Environ. Chang.* 42, 181–192 (2017).

R1.5	Regardless of the methodology to account for the trend in population, again, it is unclear how the number “4,350,912,136” people was estimated (how this aggregation was done at grid level, since the methodology to estimate trends was performed at grid level). How sure are we that this number (which is very precise in terms of digits) is not under (or over) estimating the expected affected population?	We have quantified the current and future affected population by overlaying the gridded population datasets over the wetting and drying regions with agreement greater than 50% (majority threshold) for moderate and very-high emissions showed on Figure 2 map with shades and contours for very high and intermediate emissions respectively. We have clarified it in the Methods as follow: “To estimate the global population affected, the 1km grid cells falling within the wetting and drying agreement masks are summed. To estimate the country- and state-scale affected populations, the grid cells falling within the agreement and country masks are summed.” We have rounded the population numbers to billions and millions as it is more appropriate as global scale information.	DONE
R1.6	Of particular concern is Figure 2 “warming agreement” where circles represent the affected population. Are these circles proportional to the entire population of a given country or the population effectively affected by model agreements? This needs to be properly discussed in the figure captions (bubble size shows population...).	The bubble size on Figure 2 (and Figure S3) display the affected current (and future) population by country following the approach described above on topic R1.2, rather than the entire populations. In some countries such as Bangladesh, India, Portugal and Spain, the entire population is projected to be affected by these patterns under very high emissions. We have extended the description referring to bubble size and explicitly stated “affected population” on legend title to make the figure more self-explanatory.	DONE
R1.7	c) Comments on the literature review and introduction: “Climate modes such as El Niño-Southern Oscillation, Indian Ocean Dipole, Southern Annular Mode, and Atlantic Multidecadal Oscillation contribute to precipitation variability”. The oversimplification here needs attention. It is understandable that not all possible coupled modes of variability that influence precipitation can be discussed here, but the way the sentence is written makes the non-expert reader believe that these are the only existing modes	We have edited and expanded the sentence following this suggestion, which now reads: “Unforced, internal climate fluctuations operating at timescales varying from intra-seasonal to multi-decadal additionally contribute to precipitation variability e.g. the El Niño-Southern Oscillation²², the Indian Ocean Dipole²³, the Pacific Decadal Oscillation²⁴, the Southern²⁵ and Northern²⁶ Annular Modes, and the Atlantic Multidecadal Oscillation²⁸.”	DONE

	modulating precipitation. Why the Southern Annular mode and not the Northern Annular mode, or why the PNA, the PSA, the NAO and PDO are not mentioned here? I suggest that this sentence be re-written to indicate that there are modes that vary from interannual (actually from intraseasonal) to multi-decadal time scales that influence precipitation, and cite a few of them to exemplify.		
R1.8	“Future projections for climate modes do not align over time across GCMs, resulting in highly heterogeneous projections (ref 6,7)”. This sentence seems to imply that, because climate modes (say El Nino) do not align over time across GCMs, there is disagreement among models. This is obviously not correct when considering long periods (the reason why we normally consider more than one decade). Moreover, this is not said in any of the two cited references. So, please rephrase or remove this sentence from the manuscript.	Thank you for pointing out this potential point of confusion. What we meant here was to refer to multiple climate modes acting simultaneously rather than only one. We have rewritten the sentence, which now reads: “Future projections of multiple climate modes, their interactions and resultant teleconnections with precipitation do not align over time across GCMs, resulting in highly heterogeneous projections”	
Reviewer #2			
R2.1	General Evaluation: This manuscript aims to advance our understanding of how the continuous increase of greenhouse-gases (GHG) might affect global precipitation regimes and furthermore impact global populations. An ensemble of 146 CMIP5/6 model simulations under intermediate and high emission scenarios is applied to detect agreement in wetter and drier conditions across the world and more interestingly connect precipitation change to exposed population at the country scale. There are already many regions experiencing precipitation regime change at both wet and dry end. This research indicates that this trend will continue and more and more people would be affected by the end of this century. The manuscript is well-organized, and the results are very interesting and have great significance in the field. In particular, the country-scale analysis offers a good framework for practical application such as designing/making climate adaption policies for different regions. I suggest it be accepted for publication in the current format.	Thank you very much for such fantastic feedback. We are delighted to hear your opinion about our research and very excited by the opportunity to have our manuscript under consideration for publication in Nature Communications.	NA

R2.2	Minor typos: 1) Line 227, "county" should be "country" 2) Line 244, add "is" before "worth"?	Typos have been fixed.	DONE
------	--	------------------------	------

REVIEWERS' COMMENTS

Reviewer #1 (Remarks to the Author):

Overview: this article discusses future projections of climate changing focusing on precipitation. This study investigates trends (non-parametric methods) across 146 GCMs under intermediary and elevated greenhouse emissions (models from CMIP5 and CMIP6 generations). The main goal is to provide a country-level analysis quantifying the population exposed to significant changes in precipitation regimes. The methodology considers time-series from 1980-2099 and investigates trends at model grid level. Conclusions are based on percent of agreement among models on wet or dry trends. The statistical significance is assessed by non-parametric Mann-Kendall tests, and trends are estimated with the Theil-Sen Slope to estimate the trend magnitude.

I recognize that authors made great effort in explaining their work such that misinterpretations can be avoided. The revised version of the manuscript has improved the way 'takeaway' messages are conveyed. Data, results and methods are now better presented such as key readers (scientists, media, politicians, and decision-makers) can properly use them. Caveats have been addressed, which is important when publishing articles with potential broader impacts as this one.

I think the article is ready for publication. Congratulations to authors.